



# Historical *K* index data collection of Soviet magnetic observatories, 1957–1992

Natalia Sergeyeva[1], Alexei Gvishiani[1,2], Anatoly Soloviev[1,2], Lyudmila Zabarinskaya[1], Tamara Krylova[1], Mikhail Nisilevich[1], Roman Krasnoperov[1]

[1]Geophysical Center of the Russian Academy of Sciences, Moscow, 119296, Russia
[2]Schmidt Institute of Physics of the Earth of the Russian Academy of Sciences, Moscow, 123242, Russia

*Correspondence to*: Roman Krasnoperov (r.krasnoperov@gcras.ru)

**Abstract.** *K* index is one of the oldest universal indices of geomagnetic activity, introduced in 1938 by Julius Bartels, that is still being widely used. Up to the present day, long-term timeseries of homogeneous *K* index records have been accumulated
at data repositories all over the world. The multidecadal practice of its application makes it an indispensable source of information for retrospective analysis of solar-terrestrial interaction for nearly eight Solar cycles. Most significantly, while studying the historical geomagnetic data, *K* index datasheets are in certain cases far easier for automated analysis than the conventional analogue magnetograms. The presented collection includes the results of the *K* index determination at 41 geomagnetic observatories of the former USSR for the period from July 1957 to early 1990s. This unique collection was
formed at the World Data Center for Solar-Terrestrial Physics in Moscow. The historical data, which are offered to the international scientific community, cover the second half of the 20th century and can be used for retrospective analysis and study of geomagnetic events in the past as well as for data validation or forecasting (Sergeyeva et al., 2020). The dataset is available at: https://doi.org/10.1594/PANGAEA.922233, last access: 16 September 2020.

## 1 Introduction

Preservation and providing access to historical observational data is an important and relevant problem in the Earth sciences. The World Data Center for Solar-Terrestrial Physics at the Geophysical Center of the Russian Academy of Sciences (GC RAS) consistently works on the organization of online open access to historical geomagnetic data stored in its archive. This paper presents the collection of the geomagnetic activity parameter – K index – obtained within the network of magnetic observatories and stations of the former USSR.

The most valuable part in this collection is the data of 1957–1959 related to the observation program of the International Geophysical Year (IGY), which was organized from July 1957 to December 1958, and the International Geophysical Cooperation of 1959 – the direct continuation of the IGY (Lyubovtseva et al., 2020; Nicolet, 1984).

To provide collecting, archiving and disseminating the data of all types of observations obtained within the IGY program the universal World Data Centers (WDC) were established in the USA – WDC A, and the USSR – WDC B (consisted of two
centers, B1 and B2), which received materials from the stations all over the world. The results of geomagnetic observations,



studies of the ionosphere, solar observations, data on cosmic rays and auroras were collected at WDC B2. It was established in 1956 at the Research Institute of Terrestrial Magnetism, Ionosphere and Radio Wave Propagation of the USSR Ministry of Communications (NIZMIR), which in 1959 was reassigned to the USSR Academy of Sciences (AS USSR) and renamed as the Institute of Terrestrial Magnetism, Ionosphere and Radio Wave Propagation of the AS USSR, widely known as IZMIRAN.

Later, in 1971 WDC B2 was reassigned to the Interdepartmental Geophysical Committee of the AS USSR (currently GC RAS) and designated as the World Data Center for Solar-Terrestrial Physics (WDC for STP). Currently WDC for STP is the part of the World Data System (WDS) (Raspopov et al., 2007; Rodnikov et al., 2009).

In the Soviet Union, more than 30 observatories were involved in geomagnetic research within the IGY program. The WDC for STP preserves photocopies and digital images of magnetograms, hourly mean and minute values of the Earth's magnetic
field vector components, values of indices of geomagnetic activity, data on magnetic storms and other information obtained at these observatories.

It is difficult to overestimate the importance of data from high latitude observatories established in the Arctic at various times. Among others worth to mention: Tikhaya Bay on Franz Josef Land (established in 1929 during the development of the Northern Sea Route); Dixon, Cape Wellen (Uelen) and Cape Chelyuskin (established in 1931–1933 within the Second International
Polar Year program); Heiss Island and Tixie Bay (established in 1957 within the IGY program). Magnetic observations were performed at Antarctic stations Mirny, Pionerskaya, Oasis and Vostok that were established during the First and Second Soviet Complex Antarctic Expeditions in 1956–1957. At the same period, magnetic observations were made at the North Pole (romanized 'Severny Polus') drifting stations SP-6, SP-7, and SP-8. This data was used for daily calculation of the C index (Lyubovtseva et al., 2020).

At that time, the Soviet geomagnetic observatories were equipped mainly with foreign-manufactured instruments for registration of geomagnetic variations. For example, in 1957–1959 the observatories Irkutsk, Kiev, Lvov, Tbilisi, Tashkent, Yakutsk, Vladivostok, and Mirny were equipped with the Eschenhagen variation stations and La Cour variometers. The Edelman magnetometers and Lloyd scales were installed at the Kazan, Sverdlovsk, and Odessa observatories. The Tepfer variometers were installed at the Leningrad and Vladivostok observatories. The Schultz variometer was used at the Yuzhno-
Sakhalinsk observatory. The Schmidt scales were used at the Srednikan observatory. Some observatories used domestically manufactured instruments: the Yanovsky variometer in Srednikan, the Brunelli magnetovariational station in Ashkhabad. The polar observatories in the Arctic and Antarctic (including the North Pole drifting stations) were equipped with Soviet-manufactured Brunelli magnetovariational stations and La Cour variometers. The accuracy of these instruments was low due to the influence of temperature, humidity, and other factors (Belov et al., 2006).

In the mid-1960s the re-equipment of observatories (starting with Moscow, Odessa, and Tbilisi observatories in 1964) with the new Bobrov quartz variometers began (Bobrov, 1961; Belov et al., 2006). Automatic quartz magnetovariational stations were developed in IZMIRAN for polar geomagnetic stations that conducted research in severe climatic conditions in the Arctic and Antarctic (Burtsev et al., 1977). By the mid-1980s, the network of geomagnetic observatories in the USSR was equipped with digital quartz magnetovariational stations (Belov et al., 2006).



New geomagnetic observatories that were deployed in the USSR in the 1960s–1970s participated in major international projects: the International Year of the Quiet Sun (1964–1965), the International Year of the Active Sun (1969–1971), the International Magnetospheric Study (1976–1979). The observatories continued transferring data to the WDC system (Raspopov et al., 2007).

With the collapse of the USSR in December 1991 operation of a number of observatories was terminated. Some of them
continued to transmit data to the WDC for STP until 1992.

To estimate the geomagnetic disturbances caused by extraterrestrial sources quantitatively, the indices of geomagnetic activity are widely used by the geomagnetic community (Lincoln, 1967). The majority of the indices are derived from the measurements carried out at magnetic stations and observatories (Mayaud, 1980) (Fig. 1). The International Service of Geomagnetic Indices (http://isgi.unistra.fr/, last access: 30 September 2020) handles and disseminates indices that are officially
accepted by the International Association of Geomagnetism and Aeronomy (IAGA). They are divided into regional, planetary and source driven indices and express responses from various magnetospheric-ionospheric current systems. The first and the simplest $C$ index was introduced in 1906 to estimate magnetic field activity over a day on a three-point scale (0, 1 and 2). Since that time, the index evolved into $K$ index (Bartels, 1938), which is still widely used nowadays (Gvishiani and Soloviev, 2020). During the period from 1957 to 1971, all observatories and drifting stations SP-6, SP-7 and SP-8 determined $C$ index and
transferred it to the WDCs. Since July 1957, data on geomagnetic $K$ and $C$ indices were transferred to WDC B2 by 22 observatories, and since January 1958 – by 26 observatories (Fig. 2).

Among them are the oldest observatories, established in the Russian Empire in the 19th century: Kazan, St. Petersburg (Leningrad – Voeikovo), Yekaterinburg (Sverdlovsk), Tiflis (Tbilisi), Irkutsk; and observatories established in the Soviet Union in the 20th century before the IGY (in 1930s–1940s): Yakutsk, Vladivostok, Kiev, Odessa, Lvov, Moscow, Tashkent,
Srednikan, Tomsk, Yuzhno-Sakhalinsk.

The collection of $K$ index values, stored in the WDC for STP repository, contains data from 41 observatories located on the territory of the former USSR and the Antarctic (Table 1). The collection is unique and representative because it includes data obtained over a large area covering both high, medium, and low latitudes, and consists of a long series of $K$ index values.

## 2 Description of $K$ index

Comprehensive studies of the Earth magnetic field require more frequently defined geomagnetic activity measure, devoid of any subjectivity. Therefore, in 1939 the International Association of Terrestrial Magnetism and Electricity (now International Association of Geomagnetism and Aeronomy – IAGA) introduced a three-hour index $K$ with a ten-point scale into the practice of magnetic observatories that was also proposed by Julius Bartels in 1938 (Bartels, 1938; 1939; Bartels et al., 1939). The designation $K$ derives from the German word 'Kennziffer' (Bartels, 1938), which literally means a 'characteristic parameter'
or 'identification number'. The $K$ index was designed as a measure of the range of irregular and rapid geomagnetic activity, including geomagnetic storms. It implies to be insensitive to the long-term components of magnetic variations (Love and



Remick, 2007). It is a numerical characteristic of the variability of geomagnetic activity at a given observatory over 3-hour intervals. In this particular case the variability is understood as the amplitude of the observed horizontal field relative to the quiet daily variation $Sq$. The data source for $K$ index calculating is magnetograms. Initially, when only analogue magnetograms were available, magnetologists determined the amplitudes of geomagnetic disturbances and, consequently, the $K$ index values manually (Fig. 3).

At Soviet observatories, indices were calculated once per 10 days or month based on the results of analog magnetograms' processing, following the procedure described in (Bartels et al., 1939). According to the procedure, for $K$ index calculation it is necessary to determine and remove the regular $Sq$ part of the record. Quiet daily variation $Sq$ was determined based on five daily magnetograms with clear and distinguishable characteristics of quiet days. Then amplitudes of variations of geomagnetic vector horizontal components ($H$ and $D$) for each three-hour interval were defined. Then the regular part of the quiet daily variation $Sq$ was subtracted. To facilitate this procedure, special transparent flexible gauges (transparent pallets on celluloid or other similar materials) were used. In each observatory, they were drawn using the quiet daily variation line and the observatory $K$ scale from Table 2 (Nechaev, 2006; Zabolotnaya, 2007). Considering that the amplitude of the geomagnetic disturbance depends on the latitude of the observation site (the maximum amplitude is observed in the aurora zone), the value of the $K$ scale division is selected for each observatory. The largest value of the two amplitudes obtained for each of the three-hour intervals using the observatory $K$ scale was converted to the $K$ index value and adopted as the final one. Only the horizontal $H$ and $D$ components are used to determine the $K$ index. Component $Z$ for observatories located 1,000 or 2,000 km from the auroral zone is extremely sensitive to the auroral electrojet and more than $H$ and $D$ sensitive to local effects of induced underground currents (Mayaud, 1967).

The $K$ index for a particular observatory takes values from 0 to 9 for each 3-hour interval, starting from 00:00 UT, where 9 corresponds to the strongest geomagnetic disturbances. The value $K = 9$ corresponds to a disturbance exceeding 2,500 nT in the auroral zone and about 300 nT at low latitudes (excluding the equator). For each observatory, a correspondence between the value $K = 9$ and the disturbance amplitude was obtained by considering the extremely strong geomagnetic disturbance observed on 16 April 1938. It was agreed that on this day between 06:00 and 09:00 UT the $K$ index at all observatories was equal to 9 points, and the maximum value of the disturbance amplitude over the considered interval was taken as the lower amplitude limit corresponding to $K = 9$. An event with $K \leq 2$ is quiet, $K = 2$–3 is slightly disturbed, $K = 4$ is disturbed, $K = 5$–6 is a magnetic storm, and $K \geq 7$ is a large magnetic storm.

With the advent of magnetic variometers with digital registration, data processing and calculation of $K$ index was carried out using computers. It became necessary to develop algorithms for calculating $K$ index in automatic mode. Computer programs for calculating $K$ index developed in the 1980s differ mainly in the method for determination of the quiet daily variation $Sq$ (Menvielle et al, 1995). Two main approaches are used: averaging the magnetograms of the nearest quiet days (Takahashi et al., 2001; Nechaev, 2006; Dmitriev and Filippov, 2010) and by smoothing the current magnetogram (Hopgood, 1986; Jankowski et al., 1988; Golovkov et al., 1989; Sucksdorff et al., 1991).





The ex-Soviet observatories, such as Irkutsk (IRT) or Borok (BOX), use algorithms that are as close as possible to the method of Julius Bartels that allows to preserve the continuity of the observatory $K$ index timeseries for previous years (Anisimov et al., 2015; Dmitriev and Filippov, 2010; Nechaev, 2006).

But the $K$ index is a local indicator, describing geomagnetic disturbances in the vicinity of a particular observatory. In 1949 Julius Bartels suggested a new planetary $Kp$ index for estimating the geomagnetic activity on the planetary scale over 3-hour

intervals (Bartels, 1949). The planetary $Kp$ index is calculated as the average of the $K$ indices from 13 selected observatories situated in the subauroral zones. The $K$ index values are currently used for the derivation of the IAGA planetary geomagnetic indices $Kp$ ($ap$), $am$ ($Km$), $an$ ($Kn$), $as$ ($Ks$) and $aa$ (Berthelier, 1993; Mayaud, 1980; Menvielle and Berthelier, 1991; Menvielle et al., 1995).

### 3 Compilation of the historical $K$ index data collection

Determination of the $K$ index is a standard procedure for magnetic observatory practice. The data obtained using the abovementioned procedure were prepared as monthly arrays in the form of standard datasheets. The tables for these datasheets were developed and adopted by the beginning of the IGY (Fig. 4a). The IAGA datasheet was introduced in June 1977 (Fig. 4b). In the 1980s, datasheets with addition of information on magnetic storms, recommended by IZMIRAN, was used to represent data from Soviet observatories (Fig. 5). Datasheets with $K$ index definitions were sent to the WDC B2 by mail and

stored as paper documents. Later, in the 1990s, the data exchange via e-mail started. Each datasheet indicated the value of the lower limit for $K = 9$.

To guarantee the preservation of this data and provide convenient and continuous on-line access and more efficient use, the entire array of paper datasheets was converted into digital form. The conversion was performed by scanning paper documents with transferring in the PDF format. By means of ScanSoft PaperPort software, electronic documents were edited: image crop

and alignment, removal of stains and excessive inscriptions (elements of "noise"). Where it was possible the text quality was manually improved. The following stage was visual verification of compliance of the digital version of documents with paper originals and necessary manual correction. The final structured data array in digital form (the array of digital documents) was published on the WDC for STP website with open access ([http://www.wdcb.ru/stp/data/K_indices/](http://www.wdcb.ru/stp/data/K_indices/), last access: 1 September 2020). Then the entire data set was converted into ASCII encoded text files. Most of the data represented as handwritten paper

documents was digitized manually. A small part of data was digitized using formatted text recognition and structured data extraction software, particularly the technology implemented in ABBYY FineReader software. The data files were visually checked and validated by calculating the total values in table line and comparing them to tabular daily values in Microsoft Excel 2010 software suite. Finally, all data files were converted into the uniform standard format adopted in the WDS (Fig. 6).





A detailed catalogue of data availability with monthly resolution was compiled. In total the collection includes 1054 annual files. The list of all observatories and stations that provided data for the presented collection is given in Table 1. It also shows the value of the $K = 9$ lower amplitude limits for each of the observatories.

Data for 15 observatories (ASH, CCS, CWE, DIK, HIS, IRT, KIV, KZN, LNN, LVV, MMK, MOS, TKT, YAK, YSS) represent a single long time series of observations over 34–36 years without interruption. The OAS, PIO observatories in

Antarctica, and TKH in the Arctic transferred their $K$ index data to WDC only for 2 years during the IGY period of 1957–1958. They were closed shortly after IGY ending. Unfortunately, some sets have omissions, or data was not transferred or simply lost.

**4 Applicability of FAIR principles to the historical $K$ index data set**

In the last decade, the term "FAIR Data" has become popular in scientific literature, as well as in general mass media. With

time it became clear that FAIR data can significantly strengthen the results of scientific research and lead to their substantial and, sometimes, unexpected extensions. Scientists working in geophysics and the Earth sciences in general often claim nowadays that their results are obtained using FAIR pieces of information.

Following (https://www.go-fair.org/fair-principles/, last access: 1 September 2020) we, hereafter, shortly present the FAIR data formalism simultaneously showing that the historical $K$ index data collection presented in this article fully satisfies the

FAIR conditions.

The abbreviation FAIR stands for 'Findable', 'Accessible', 'Interoperable' and 'Reusable' data sets. Hereafter are the most valuable clarifications of what each of these four features means (Wilkinson et al., 2016). Along with that we show how the requirements are satisfied for the data set of the $K$ indices under consideration (*in italic*).

**To be Findable:**

F1. data are assigned a unique and eternally persistent identifier (*see the DOI of the K index collection at PANGAEA data repository*).

F2. data are described with rich metadata (*provided in the PANGAEA system along with a comprehensive description given in this paper*).

F3. data are registered and indexed in a searchable resource (*in the PANGAEA data repository itself*).

**To be Accessible:**

A1. data are retrievable by their identifier using a standardized communications protocol (*implemented by DataCite system, which ensures adequate DOI retrievals*).

A2. the protocol is open, free, and universally implementable (*the DataCite system fully satisfies this requirement*).

A3. metadata are accessible, even when the data are no longer available (*this is provided by independent storage of the*

*metadata in the DataCite system*).

**To be Interoperable:**



I1. data and metadata use a formal and broadly applicable means for knowledge representation (*this is ensured by using the English language and standard Excel tables for data storage*).

I2. data and metadata include qualified references from other resources (*information description in this paper includes sufficient references on the dataset*).

**To be Reusable:**

R1. data have accurate and relevant attributes (*in the PANGAEA system the data are attributed with abstract, keywords, geographic coverage, dataset size, etc.*).

R2. (meta)data are released with a clear and accessible data usage license (*information on data usage license is provided by the PANGAEA system*).

R3. (meta)data are associated with their provenance (*information on the provenance of the K index collection is given in detail in this paper as well as in the dataset description in the PANGAEA system*).

R4. (meta)data meet domain-relevant community standards (*the presented K index collection fully meets the requirements of IAGA and WDS*).

Hence, the FAIR requirements for the considered historical *K* index data collection are fully satisfied.

**5 Conclusion**

Implementation of digital magnetometers and data loggers that provide near real-time transfer of observational information to geophysical data repositories has become a pivotal moment in the observatory practice. But fast and convenient access to digital magnetograms from observatories all over the world did not completely substitute the traditional parameters of geomagnetic activity. *K* index – the classical commonly used parameter of geomagnetic activity – serves as the measure of local magnetic field variations within three-hour intervals. The *K* index values defined at different observatories are used for calculating other geomagnetic activity indices such as *Kp* (*ap*), *am* (*Km*), *an* (*Kn*), *as* (*Ks*) and *aa*. The *K* index is used for studying of magnetic storms – the most vivid reflection of the Sun's impact on the Earth's magnetosphere. The study of magnetic storms and geomagnetic variations that they induce is important for analyzing their impact on meteorological processes, biosphere, technological infrastructure, etc.

The long-time practice of the *K* index application makes it an important instrument for retrospective complex analysis of solar-terrestrial interaction. The beginning of coordinated large-scale geomagnetic observations in the USSR was initiated during the International Geophysical Year of 1957–1958. Although some of the observatories were temporary and were discontinued by the end of 1958, the observational network of the former USSR remained relatively dense and provided the sufficient spatial coverage. The World Data Center (WDC) B2 which later was renamed as WDC for Solar-Terrestrial Physics (WDC for STP) in Moscow became the main repository of data on geomagnetism in the former USSR. The *K* index datasheets were transferred from observatories to the WDC for STP where they are stored as paper documents. In 2000s and 2010s WDC for STP initiated an ambitious project for digitizing this archive of paper documents. The presented unique collection of historical *K* index



values (Sergeyeva et al., 2020) is the result of this activity. It is a vast massif of homogeneous data that cover more than three
decades of geomagnetic measurements at observatories of the former USSR. Considering the FAIR principles in regard to the
presented data collection, in this article we come up with a new example of FAIR data.

## 6 Data availability

The data on the *K* index from paper catalogues was digitized at Geophysical Center of RAS by the team of the World Data
Center for Solar-Terrestrial Physics. Digital data are available at: https://doi.org/10.1594/PANGAEA.922233, last access: 16
September 2020 (Sergeyeva et al., 2020). Photocopies of the datasheets and analogue magnetograms from geomagnetic
observatories of the former USSR since 1957 are accessible via the web-page of the World Data Center for Solar-Terrestrial
Physics: http://www.wdcb.ru/stp/data.html, last access: 4 September 2020.

**Author contributions.** NS, AG, AS, and RK – preparation of the manuscript and figures. LZ, TK, and MN – digitalization of
analogue data, data validation, and preparation of the *K* index data collection. MN – publication of data in the PANGAEA
system.

**Competing interests.** The authors declare that they have no conflict of interest.

**Acknowledgements.** The authors wish to thank the team of the World Data Center for Solar-Terrestrial Physics in Moscow,
Russia, for preservation and making publicly available historical geophysical data. This work employed data provided by the
Shared Research Facility "Analytical Geomagnetic Data Center" of the Geophysical Center of RAS (http://ckp.gcras.ru/, last
access: 1 September 2020).

**Financial Support.** This work was conducted in the framework of budgetary funding of the Geophysical Center of RAS,
adopted by the Ministry of Science and Higher Education of the Russian Federation.

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





305  **Tables**

**Table 1: Geomagnetic observatories of the former USSR. K index data availability**

| No. | Code | Name of the observatory | Country | Geographical coordinates | | Altitude, m | Lower limit for $K = 9$, nT | $K$ index Availability of data in the form of digital files, years |
|---|---|---|---|---|---|---|---|---|
| | | | | Latitude | Longitude | | | |
| 1 | AAA | Alma- Ata | Kazakhstan | 43.250 | 76.920 | 1300 | 300 | 1964–1981, 1983–1992 |
| 2 | ARK | Arkhangelsk | Russia | 64.583 | 40.5000 | | 300 | 1991–1992 |
| 3 | ARS | Arti (Yekaterinburg) | Russia | 56.433 | 58.567 | 290 | 550 | 1976–1992 |
| 4 | ASH | Ashkhabad (Vannovskaya) | Turkmenistan | 37.950 | 58.100 | 300 | 300 | 1958–1991 |
| 5 | CCS | Cape Chelyuskin | Russia | 77.717 | 104.283 | 10 | 2500 | 1957–1992 |
| 6 | CWE | Cape Wellen (Uelen) | Russia | 66.170 | −169.830 | 10 | 1250 | 1957–1992 |
| 7 | DIK | Dikson (Dixon) | Russia | 73.543 | 80.562 | 20 | 1500 | 1957–1992 |
| 8 | HIS | Heiss Island (Druzhnaya) | Russia | 80.620 | 58.050 | 20 | 2000 | 1957–1992 |
| 9 | IRT | Irkutsk (Patrony) | Russia | 52.167 | 104.450 | 460 | 350 | 1957–1992 |
| 10 | KGD | Karaganda (Bereznyaki) | Kazakhstan | 49.820 | 73.080 | | 350 | 1973–1981, 1983–1990 |
| 11 | KIV | Kiev (Dymer) | Ukraine | 50.720 | 30.300 | 100 | 350 | 1958–1991 |
| 12 | KNG | Kaliningrad | Russia | 54.600 | 20.200 | | 600 | 1977–1981, 1983–1992 |
| 13 | KZN | Kazan (Zaymishe) | Russia | 55.830 | 48.850 | 80 | 550 | 1957–1992 |
| 14 | LNN | Leningrad (Voyeykovo) | Russia | 59.950 | 30.700 | 70 | 600 | 1957–1992 |
| 15 | LVV | Lvov | Ukraine | 49.900 | 23.750 | 400 | 550 | 1957–1991 |
| 16 | MGD | Magadan (Stekol'nyj) | Russia | 60.120 | 151.020 | | 550 | 1965–1972, 1976–1994 |
| 17 | MIR | Mirny | Antarctica (Russia) | −66.550 | 93.020 | 20 | 2000 | 1957–1981, 1983-1990 |
| 18 | MMK | Loparskaya (Murmansk) | Russia | 68.250 | 33.080 | 210 | 2500 | 1957–1993 |
| 19 | MNK | Minsk (Pleshchenicy) | Belorussia | 54.100 | 26.520 | | 550 | 1962–1981, 1983–1991 |
| 20 | MOL | Molodezhnaya | Antarctica (Russia) | −67.670 | 45.850 | 0 | 2000 | 1975–1981, 1983-1990 |
| 21 | MOS | Moscow (Krasnaya Pakhra) | Russia | 55.467 | 37.312 | 190 | 550 | 1949–1992 |
| 22 | NKK | Novokazalinsk | Kazakhstan | 45.770 | 62.120 | | 350 | 1973–1981, 1983–1990 |
| 23 | NVL | Novolazarevskaya | Antarctica (Russia) | −70.770 | 11.820 | 460 | 1500 | 1973–1981, 1983-1988 |
| 24 | NVS | Novosibirsk | Russia | 55.030 | 82.900 | | 500 | 1969–1992 |
| 25 | OAS | Oasis | Antarctica (Russia) | −66.300 | 100.720 | | 2000 | 1957–1958 |
| 26 | ODE | Odessa (Stepanovka) | Ukraine | 46.780 | 30.880 | 140 | 350 | 1957–1981, 1983–1991 |
| 27 | PET | Petropavlovsk (Paratunka) | Russia | 53.100 | 158.630 | | 450 | 1969–1971, 1973–1992 |
| 28 | PIO | Pionerskaya | Antarctica (Russia) | −69.730 | 95.500 | | 2000 | 1957–1958 |
| 29 | POD | Podkamennaya Tunguska | Russia | 61.400 | 90.000 | | 650 | 1969–1971, 1974–1992 |
| 30 | SRE | Srednikan | Russia | 62.430 | 152.320 | | 550 | 1957–1966 |



| 31 | SVD | Sverdlovsk (Vysokaya Dubrava) | Russia | 56.730 | 61.070 | 290 | 550 | 1957–1977 |
|---|---|---|---|---|---|---|---|---|
| 32 | TFS | Tbilisi (Dusheti) | Georgia | 42.080 | 44.700 | 982 | 350 | 1957–1991 |
| 33 | TIK | Tixie Bay (Tiksy) | Russia | 71.580 | 129.000 | 40 | 1000 | 1957–1967, 1969–1991 |
| 34 | TKH | Tikhaya Bay | Russia | 80.300 | 52.800 | | 2000 | 1957–1958 |
| 35 | TKT | Tashkent (Yangi Bazar) | Uzbekistan | 41.333 | 69.617 | 500 | 300 | 1957–1991 |
| 36 | TMK | Tomsk | Russia | 56.470 | 84.930 | 200 | 350 | 1958–1970 |
| 37 | UBA | Ulan–Bator | Mongolia | 47.850 | 106.750 | | 300 | 1973–1975, 1978-1980, 1983-1989 |
| 38 | VLA | Vladivostok (Gornotayozhnaya) | Russia | 43.697 | 132.160 | 178 | 300 | 1957–1981, 1983-1988,1990 |
| 39 | VOS | Vostok | Antarctica (Russia) | −78.450 | 106.867 | 3500 | 2000 | 1958–1981, 1983-1990 |
| 40 | YAK | Yakutsk | Russia | 62.020 | 129.720 | 100 | 550 | 1957–1991 |
| 41 | YSS | Yuzhno Sakhalinsk | Russia | 46.950 | 142.717 | 70 | 350 | 1957–1990 |

**Table 2: Correspondence of geomagnetic disturbance amplitudes in nT to K index values depending on the geomagnetic latitude of observatory.**

| Scale | $K$ index | | | | | | | | | | Lat |
|---|---|---|---|---|---|---|---|---|---|---|---|
| | 0 | 1 | 2 | 3 | 4 | 5 | 6 | 7 | 8 | 9 | |
| 1 | 0–25 | 25–50 | 50–100 | 100–200 | 200–350 | 350–600 | 600–1000 | 1000-1650 | 1650–2500 | >2500 | 64–90 |
| 2 | 0–20 | 20–40 | 40–80 | 80–160 | 160–280 | 280–480 | 480–800 | 800–1300 | 1300–2000 | >2000 | 65–80 |
| 3 | 0–18 | 18–36 | 36–72 | 72–144 | 144–252 | 252–432 | 432–720 | 720–1188 | 1188–1800 | >1800 | 80 |
| 4 | 0–15 | 15–30 | 30–60 | 60–120 | 120–210 | 210–360 | 360–600 | 600–1000 | 1000–1500 | >1500 | 60–83 |
| 5 | 0–12 | 12–25 | 25–50 | 50–100 | 100–175 | 175–300 | 300–500 | 500–825 | 825–1250 | >1250 | 62 |
| 6 | 0–10 | 10–20 | 20–40 | 40–80 | 80–140 | 140–240 | 240–400 | 400–660 | 660–1000 | >1000 | 60–83 |
| 7 | 0–8 | 8–15 | 15–30 | 30–60 | 60–105 | 105–180 | 180–300 | 300–500 | 500–750 | >750 | 58–62 |
| 8 | 0–6 | 6–12 | 12–24 | 24–48 | 48–85 | 85–145 | 145–240 | 240–400 | 400–600 | >600 | 55–58 |
| 9 | 0–5 | 5–10 | 10–20 | 20–40 | 40–70 | 70–120 | 120–200 | 200–330 | 330–550 | >550 | 48–54 |
| 10 | 0–5 | 5–10 | 10–20 | 20–40 | 40–70 | 70–120 | 120–200 | 200–330 | 330–550 | >500 | 36–57 |
| 11 | 0–4 | 4–8 | 8–16 | 16–30 | 30–50 | 50–85 | 85–140 | 140–230 | 230–350 | >350 | 31–47 |
| 12 | 0–3 | 3–6 | 6–12 | 12–24 | 24–40 | 40–70 | 70–120 | 120–220 | 200–300 | >300 | 0–40 |

310





# Figures

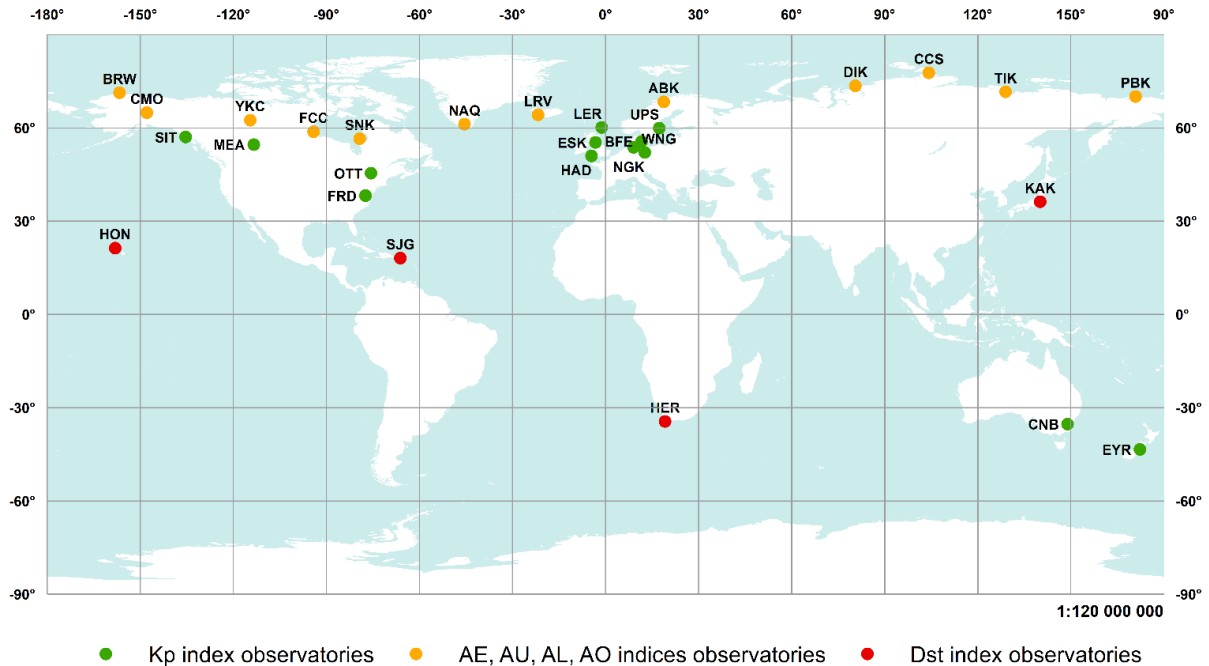

**Figure 1: The map shows the observatories used for calculating the geomagnetic indices:** *Kp* **(green circles);** *AE, AU, AL, AO* **(orange circles) and** *Dst* **(red circles).**

**Figure 2: Location of Soviet geomagnetic observatories that provided data for the presented collection: a) continental territory of the USSR; b) the Antarctic.**

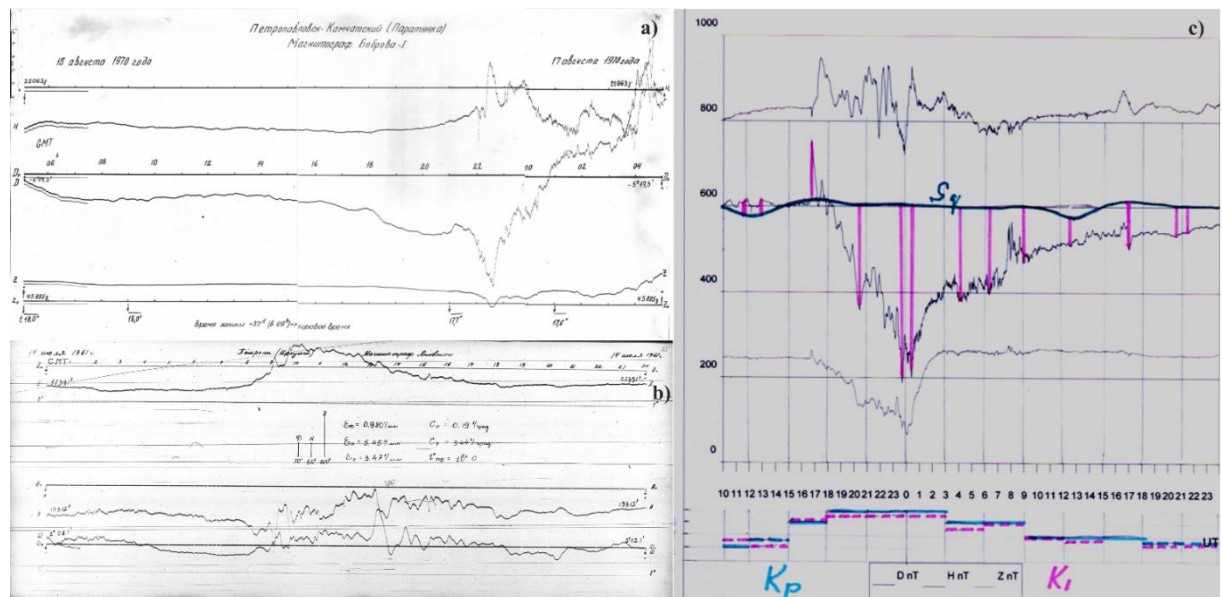

320

**Figure 3: Photocopies of analogue magnetograms: a) Petropavlovsk (Paratunka) observatory, 16—17.08.1970, Bobrov variometer; Irkutsk (Patrony), 14.07.1960, Yanovsky variometer; c) manual determination of *K* (purple bars) and *Kp* values (http://www.wdcb.ru/stp/index.en.html, last access: 1 September 2020).**



**Figure 4: Paper datasheets for _K_ index values for observatory Irkutsk (Patrony): a) datasheet form adopted for the IGY (July 1957); b) datasheet form adopted by IAGA in 1977 (July 1977) (http://www.wdcb.ru/stp/data/K_indices/IRT_Irkutsk/, last access: 1 September 2020).**

a) IRKUTSK — Иркутск — Суточные характеристики — Июль 1957г.

| Дата | ΔD | ΔH | ΔZ | Характер C | 0–3 | 3–6 | 6–9 | 9–12 | 12–15 | 15–18 | 18–21 | 21–24 | ΣК | Дата |
|---|---|---|---|---|---|---|---|---|---|---|---|---|---|---|
| 1 | | | | 2 | 4 | 4 | 5 | 3 | 2 | 3 | 6 | 3 | 30 | 1 |
| 2 | | | | 2 | 2 | 1 | 3 | 4 | 6 | 5 | 3 | 2 | 26 | 2 |
| 3 | | | | 1 | 4 | 4 | 4 | 4 | 5 | 1 | 2 | 2 | 26 | 3 |
| 4 | | | | 0 | 2 | 2 | 2 | 2 | 1 | 1 | 2 | 4 | 16 | 4 |
| 5 | | | | 1 | 5 | 6 | 6 | 5 | 2 | 1 | 2 | 3 | 30 | 5 |
| 6 | | | | 0 | 3 | 3 | 3 | 3 | 2 | 2 | 3 | 3 | 22 | 6 |
| 7 | | | | 0 | 3 | 3 | 2 | 3 | 2 | 1 | 2 | 2 | 18 | 7 |
| 8 | | | | 0 | 2 | 3 | 2 | 2 | 2 | 1 | 2 | 2 | 16 | 8 |
| 9 | | | | 0 | 2 | 2 | 2 | 2 | 1 | 2 | 1 | 1 | 13 | 9 |
| 10 | | | | 0 | 2 | 2 | 2 | 2 | 0 | 0 | 1 | 1 | 10 | 10 |
| 11 | | | | 0 | ~ | 1 | 2 | 2 | 4 | 1 | 1 | 1 | — | 11 |
| 12 | | | | 0 | 3 | 2 | 3 | 2 | 2 | 2 | 2 | 2 | 18 | 12 |
| 13 | | | | 0 | 1 | 2 | 2 | 2 | 0 | 1 | 1 | 2 | 11 | 13 |
| 14 | | | | 0 | 2 | 2 | 2 | 2 | 2 | 1 | 2 | 3 | 16 | 14 |
| 15 | | | | 0 | 3 | 2 | 1 | 2 | 1 | 0 | 0 | 2 | 11 | 15 |
| 16 | | | | 1 | 3 | 2 | 4 | 3 | 3 | 3 | 2 | 3 | 23 | 16 |
| 17 | | | | 0 | 3 | 3 | 2 | 3 | 2 | 2 | 2 | 3 | 20 | 17 |
| 18 | | | | 0 | 2 | 2 | 3 | 3 | 2 | 3 | 2 | 3 | 20 | 18 |
| 19 | | | | 1 | 2 | 4 | 4 | 3 | 3 | 4 | 3 | 3 | 26 | 19 |
| 20 | | | | 0 | 2 | 3 | 2 | 3 | 2 | 1 | 3 | 2 | 18 | 20 |
| 21 | | | | 0 | 2 | 2 | 2 | 3 | 1 | 1 | 1 | 2 | 14 | 21 |
| 22 | | | | 1 | 3 | 3 | 4 | 4 | 3 | 3 | 4 | 3 | 27 | 22 |
| 23 | | | | 1 | 3 | 4 | 3 | 1 | 1 | 1 | 1 | 3 | 17 | 23 |
| 24 | | | | 0 | 2 | 3 | 3 | 4 | 3 | 3 | 2 | 2 | 22 | 24 |
| 25 | | | | 0 | 2 | 2 | 3 | 1 | 2 | 3 | 2 | ~ | — | 25 |
| 26 | | | | 0 | 1 | 1 | 1 | 0 | 2 | 1 | 1 | 2 | 09 | 26 |
| 27 | | | | 0 | 2 | 1 | 1 | 2 | 1 | 2 | 3 | 3 | 15 | 27 |
| 28 | | | | 0 | 3 | 2 | 2 | 3 | 1 | 1 | 1 | 2 | 15 | 28 |
| 29 | | | | 1 | 3 | 3 | 4 | 4 | 3 | 3 | 1 | 1 | 22 | 29 |
| 30 | | | | 0 | 2 | 2 | 3 | 3 | 1 | 1 | 1 | 1 | 14 | 30 |
| 31 | | | | 0 | 3 | 3 | 2 | 2 | 1 | 1 | 2 | 3 | 17 | 31 |
| Среднее | $\delta D = 5.54$ | $\delta H = 4.57$ | $\delta Z = 3.28$ | | | | | $Kg = 350$ | | | | | | |

Зак. № 1093  2000

b) INTERNATIONAL UNION OF GEODESY AND GEOPHYSICS — INTERNATIONAL ASSOCIATION OF GEOMAGNETISM AND AERONOMY

INDICES OF GEOMAGNETIC ACTIVITY

OBSERVATORY Patrony (Irkutsk)  SCALE-VALUES OF VARIOMETERS
MONTH July 19 77  IN γ/MM
RANGE FOR K=9  350 γ  D....... H....... Z.......

| GR DAY | 00h–03h | 03h–06h | 06h–09h | 09h–12h | 12h–15h | 15h–18h | 18h–21h | 21h–24h | SUM |
|---|---|---|---|---|---|---|---|---|---|
| 1 | 3 | 0 | 0 | 5 | 2 | 4 | 3 | 3 | 20 |
| 2 | 3 | 3 | 3 | 2 | 2 | 2 | 3 | 4 | 22 |
| 3 | 3 | 2 | 4 | 3 | 3 | 2 | 2 | 2 | 21 |
| 4 | - | - | - | - | - | - | - | - | - |
| 5 | 2 | 1 | 3 | 2 | 3 | 2 | 3 | 1 | 17 |
| 6 | 3 | 3 | 4 | 4 | 6 | 5 | 4 | 3 | 32 |
| 7 | 4 | 3 | 4 | 3 | 3 | 3 | 2 | 3 | 25 |
| 8 | 3 | 3 | 3 | 4 | 3 | 3 | 2 | 3 | 24 |
| 9 | 3 | 3 | 3 | 4 | 2 | 2 | 2 | 4 | 23 |
| 10 | 3 | 3 | 3 | 3 | 4 | 3 | 3 | 3 | 25 |
| 11 | 3 | 3 | 4 | 2 | 3 | 2 | 3 | 3 | 23 |
| 12 | 2 | 2 | 2 | 2 | 0 | 0 | 0 | 0 | 08 |
| 13 | 2 | 3 | 3 | 2 | 3 | 2 | 3 | 3 | 21 |
| 14 | 4 | 5 | 5 | 4 | 3 | 2 | 2 | 3 | 28 |
| 15 | 2 | 2 | 3 | 2 | 2 | 2 | 2 | 4 | 19 |
| 16 | 2 | 4 | 4 | - | - | - | - | - | 10 |
| 17 | 2 | 3 | 4 | 3 | 2 | 3 | 3 | 4 | 24 |
| 18 | 2 | 1 | 2 | 3 | 3 | 2 | 2 | 4 | 19 |
| 19 | 2 | 3 | 5 | 4 | 4 | 4 | 3 | 4 | 29 |
| 20 | 3 | 4 | 4 | 4 | 5 | 3 | 3 | 4 | 30 |
| 21 | 3 | 2 | 2 | 2 | 2 | 3 | 4 | 0 | 18 |
| 22 | 2 | 2 | 3 | 3 | - | - | - | - | 10 |
| 23 | 2 | 2 | 3 | 3 | 2 | 3 | 3 | 2 | 20 |
| 24 | 2 | 2 | 2 | 2 | 2 | 1 | 2 | 2 | 15 |
| 25 | 2 | 3 | 3 | 2 | 1 | 2 | 1 | 3 | 17 |
| 26 | 0 | 2 | 2 | 2 | 1 | 2 | 1 | 3 | 13 |
| 27 | 1 | 1 | 1 | 1 | 1 | 1 | 1 | 2 | 09 |
| 28 | 1 | 1 | 2 | 2 | 1 | 2 | 1 | 3 | 13 |
| 29 | 5 | 6 | 5 | 5 | 7 | 5 | 2 | 3 | 38 |
| 30 | 5 | 3 | 4 | 4 | 2 | 2 | 2 | 2 | 24 |
| 31 | 2 | 2 | 2 | 3 | 2 | 2 | 2 | 3 | 18 |

Form No. 1082.





МЕСЯЧНЫЙ ОБЗОР СОСТОЯНИЯ МАГНИТНОГО ПОЛЯ

месяц *январь* 1990г.                 ОБСЕРВАТОРИЯ *Душети (Тбилиси*
время мировое UT                      ведомство *Инст. Геофизики*
                                              *АН Груз. ССР*

| UT / дни | 0-3 | 3-6 | 6-9 | 9-12 | 12-15 | 15-18 | 18-21 | 21-24 | ΣH | № бури дата | Начало дата | UT | Конец дата | UT | про дол. | Ампл. D | Ампл. H | Ампл. Z | характ. бури | Начало дата | UT | Конец дата | UT |
|---|---|---|---|---|---|---|---|---|---|---|---|---|---|---|---|---|---|---|---|---|---|---|---|
| 1 | 3 | 3 | 3 | 3 | 3 | 4 | 5 | 4 | 28 | | 30.I | 7:00 | 31.I | 03:05 | 20 | 92 | 87 | 44 | M | 30.I | 07:00 | 30.I | 09:00 |
| 2 | 4 | 2 | 3 | 3 | 3 | 2 | 3 | 4 | 24 | | | | | | | | | | | 30.I | 13:00 | 30.I | 15:00 |
| 3 | 4 | 3 | 2 | 3 | 3 | 4 | 3 | 3 | 25 | | | | | | | | | | | 30.I | 17:00 | 30.I | 23:00 |
| 4 | 2 | 2 | 3 | 4 | 3 | 2 | 2 | 3 | 21 | | | | | | | | | | | 31.I | 01:00 | 31.I | 03:00 |
| 5 | 3 | 2 | 3 | 3 | 3 | 4 | 4 | 4 | 26 | | | | | | | | | | | | | | |
| ⑥ | 3 | 3 | 2 | 2 | 2 | 2 | 2 | 2 | 18 | | | | | | | | | | | | | | |
| ⑦ | 2 | 2 | 2 | 2 | 2 | 1 | 1 | 2 | 14 | | | | | | | | | | | | | | |
| 8 | 2 | 2 | 2 | 2 | 4 | 5 | 4 | 4 | 25 | | | | | | | | | | | | | | |
| 9 | 4 | 3 | 3 | 3 | 3 | 2 | 2 | 4 | 24 | | | | | | | | | | | | | | |
| 10 | 4 | 2 | 3 | 4 | 4 | 4 | 4 | 3 | 28 | | | | | | | | | | | | | | |
| 11 | 3 | 2 | 3 | 3 | 3 | 4 | 4 | 3 | 25 | | | | | | | | | | | | | | |
| 12 | 3 | 3 | 3 | 4 | 4 | 3 | 4 | 2 | 26 | | | | | | | | | | | | | | |
| 13 | 3 | 2 | 2 | 3 | 2 | 4 | 2 | 2 | 20 | | | | | | | | | | | | | | |
| 14 | 2 | 3 | 2 | 3 | 3 | 3 | 2 | 2 | 20 | | | | | | | | | | | | | | |
| 15 | 2 | 3 | 2 | 3 | 3 | 4 | 2 | 2 | 21 | | | | | | | | | | | | | | |
| 16 | 2 | 3 | 2 | 3 | 2 | 4 | 4 | 4 | 24 | *Малая магнитная буря 30°-31°* | | | | | | | | | | | | | |
| ⑰ | 3 | 2 | 2 | 2 | 3 | 2 | 1 | 2 | 17 | *имеет постепенное начало.* | | | | | | | | | | | | | |
| 18 | 3 | 2 | 3 | 3 | 3 | 3 | 2 | 3 | 22 | | | | | | | | | | | | | | |
| 19 | 2 | 2 | 1 | 2 | 2 | 3 | 3 | 3 | 18 | | | | | | | | | | | | | | |
| 20 | 3 | 3 | 3 | 3 | 3 | 5 | 4 | 3 | 27 | | | | | | | | | | | | | | |
| 21 | 4 | 3 | 3 | 3 | 4 | 4 | 4 | 3 | 28 | | | | | | | | | | | | | | |
| 22 | 4 | 3 | 3 | 4 | 3 | 4 | 3 | 3 | 27 | | | | | | | | | | | | | | |
| 23 | 3 | 3 | 3 | 4 | 3 | 3 | 5 | 3 | 27 | | | | | | | | | | | | | | |
| 24 | 4 | 4 | 4 | 4 | 3 | 4 | 5 | 5 | 33 | | | | | | | | | | | | | | |
| 25 | 5 | 3 | 3 | 4 | 4 | 3 | 2 | 2 | 26 | | | | | | | | | | | | | | |
| 26 | 2 | 3 | 2 | 4 | 4 | 3 | 2 | 2 | 22 | | | | | | | | | | | | | | |
| ㉗ | 2 | 3 | 1 | 2 | 1 | 1 | 2 | 1 | 13 | | | | | | | | | | | | | | |
| 28 | 1 | 2 | 2 | 3 | 3 | 2 | 4 | 3 | 20 | Обзор составил ______ | | | | | | | | | | | | | |
| 29 | 4 | 3 | 4 | 5 | 4 | 3 | 4 | 4 | 31 | | | | | | | | | | | | | | |
| 30 | 4 | 4 | 3 | 3 | 4 | 4 | 5 | 3 | 30 | | | | | | | | | | | | | | |
| 31 | 4 | 3 | 3 | 3 | 4 | 4 | 3 | 2 | 26 | | | | | | | | | | | | | | |

**Figure 5: Monthly *K* index values datasheet for observatory Tbilisi (TFS), Georgia (January 1990), compiled according to the form,**
**proposed by IZMIRAN (http://www.wdcb.ru/stp/data/K_indices/TFS_Tbilisi/TFS_1990_1-7+st.pdf, last access: 1 September 2020).**



Line 1:   from the 9th position - Observatory name and field value for k = 9 in nT.
Line 2:   1 - 3 positions - Observatory code.
Line 3:   1 -  4  i4    Year.
   6 -  7  i2    Month.
   9 - 10  i2    Day.
   13 - 15  i3    Day of the year.
   17 - 32  8i2   Eight 3-hourly values of the index for the day.
   35 - 36  i2    The sum of 8 values of the index per day.
Line 4 and all subsequent to the end of the month:
   6 -  7  i2    Month.
   9 - 10  i2    Day.
   13 - 15  i3    Day of the year.
   17 - 32  8i2   Eight 3-hourly values of the index for the day.
   35 - 36  i2    The sum of 8 values of the index per day.
Then each new month begins with the 2nd line.
A missing value is identified by "-1".

**Figure 6: Format description for *K* index data (http://www.wdcb.ru/stp/geomag/format_K_index.html, last access: 1 September 2020).**