# Peer review of "Historical *K* index data collection of Soviet magnetic observatories, 1957–1992"

_Earth System Science Data, 2020_

## Referee Comment (RC1) · Anonymous Referee #1 · 22 Oct 2020

The manuscript 'Historical K index data collection of Soviet magnetic observatories, 1957–1992' by Natalia Sergeyeva and others, submitted to ESSD, https://doi.org/10.5194/essd-2020-270, describes the geomagnetic K indices collected at a large number of geomagnetic observatories in the second half of the 20th century. This data set is scientifically valuable and its publication is laudable. It is a relevant subject for ESSD. The description of the data set, which I review here, is well written and I have only minor comments to the authors.

Comment to Editor:

line 185 to 189 I do not know enough about DataCite so I am not sure if it is relevant here. This point should be covered by you or another reviewer

[Figure]

Comments to authors:

line 11: eight Solar cycles -> eight solar cycles?

line 23: K should be italics

line 48: Please explain the term Severny Polus (or Severnyy Polyus)? Also, I am not sure if the term 'romanized' is well known by many readers. How about 'transcribed into the Latin alphabet' ?

line 71: extraterrestrial -> solar wind and magnetosphere-ionosphere interaction?

line 98: relative to -> minus?

line 118 to 122: I am not aware of the significance of the geomagnetic disturbance of April 16, 1938. Can you give references for this information?

line 122 to 123: Can you give a reference for this classification of events?

line 144: definitions -> determinations?

line 193: Data is stored not in Excel but in ASCII-Text-files?

Figure 3 caption: Please mention in the figure caption the quiet curve shown in panel (c)
* * *

---

## Referee Comment (RC2) · Anonymous Referee #2 · 23 Jan 2021

Referee's report on "Historical K index data collection of Soviet magnetic observatories, 1957–1992" by Sergeyeva et al. (#Eessd-2020-270)

Date of the comments: January 23, 2021

This manuscript describes the history of Soviet geomagnetic observatories and data centers, general introduction of the K index, and historical K index data collection that are digitized from paper datasheets. The historical K index data collection is implemented as it follows the FAIR principles. The referee thinks that the manuscript is well-written and the data collection is very important. The manuscript is worth publishing in "Earth System Science Data" after the authors make minor revision according to the following comments.

[Figure]

1. Lines 80–81 and Figure 2 The text describes 22 and 26 observatories, although Figure 2 shows locations of 41 geomagnetic observatories that may correspond to Table 1. Please indicate locations of the 22 and 26 observatories in Figure 2 or move the statement of "(Fig. 2)" to Line 87.

2. Lines 99–101 and Figure 3 (a) This line is too simple to understand Figure 3. Needs more detailed statements. (b) Figures 3a and 3b are too small to read labels or necessary information. Please enlarge. In figure caption, "b)" is missed. (c) In Figure 3c, the purple bars are drawn only for the second trace (probably the H component), but why? The K index should be determined from both the H and D components (Lines 105–106). (d) What is the date of the magnetograms in Figure 3c?

3. Table 2 Scales 3 and 7 are not adopted to the 41 geomagnetic observatories shown in Table 1. Are these scales needed in this table?

4. Line 145–146 and Figure 5 In Figure 5, there seems no indication of K9 limit.

5. Lines 163–164 The long-term data are very important. The referee suggests showing plots of the long-term (34–36 year) K index from a few representative observatories. Such plots will interest readers to access the data collection.

---

## Author Comment (AC1) · 19 Feb 2021

Thre authors would like to express their gratitude to the Referee for his or her comments, that helped us to improve the manuscript.

Line 11: eight Solar cycles -> eight solar cycles? We agree with the reviewer. In this case, there must be a lowercase letter: eight solar cycles.

Line 23: K should be italics. We agree with the reviewer. K should be written in italics.

Line 48: Please explain the term Severny Polus (or Severnyy Polyus)? Also, I am not sure if the term 'romanized' is well known by many readers. How about 'transcribed into the Latin alphabet'? There are several variants of transliteration of Cyrillic letters in Latin: 'Severniy Polyus', 'Severnyiy Polyus', or 'Severnyj Polyus'. We have replaced

the transliteration version in the text with 'Severnyj Polyus' and the abbreviations SP to NP. We have also replaced the term 'romanized' with 'transcribed Cyrillic to Latin'.

Line 71: extraterrestrial -> solar wind and magnetosphere-ionosphere interaction? We agree that it is better to reveal the term extraterrestrial and describe in more detail the mechanism causing disturbances of the geomagnetic field, which is characterized by the indices of geomagnetic activity. We have replaced the sentence "To estimate the geomagnetic disturbances caused by extraterrestrial sources quantitatively, the indices of geomagnetic activity are widely used by the geomagnetic community (Lincoln, 1967)." with "To quantify geomagnetic disturbances caused by the interaction of solar corpuscular radiation with the magnetosphere, processes in the magnetosphere itself, the interaction of the magnetosphere and the ionosphere, as well as processes in the ionosphere itself, the geomagnetic community widely uses geomagnetic activity indices (Lincoln, 1967)."

Line 98: relative to -> minus? We agree with the reviewer. We have replaced relative to with minus. In this particular case the variability is understood as the amplitude of the observed horizontal field minus the quiet daily variation Sq.

Lines 118 to 122: I am not aware of the significance of the geomagnetic disturbance of April 16, 1938. Can you give references for this information? We have added the following text to the article (line 125–127): "This fact is described in articles (Bartels et al., 1939; Lincoln, 1967). For observatories established after 1938, the lower limit of the amplitude for K = 9 is chosen in consultation with the working group on geomagnetic activity indices IAGA (before 1954 IATME) (Lincoln, 1967)."

Lines 122 to 123: Can you give a reference for this classification of events? We have added the following text (lines 127–132): "An event with K $\leq$ 2 is quiet, K = 2–3 is slightly disturbed, K = 4 is disturbed, K = 5–6 is a magnetic storm, and K $\geq$ 7 is a large magnetic storm. This version of the classification is published on the Paratunka Observatory website http://www.ikir.ru/ru/Departments/Paratunka/lfg/txt/k-index-doc.html.

Another classification is given in the work (Menviell et al., 2011): "The modern consensus is that K = 0–2 correspond to periods of magnetic quietness; K = 3–5 correspond to periods of moderate geomagnetic activity; K =6–9 correspond to periods of intense to very intense geomagnetic activity". We added the corresponding reference (line 296): Menvielle, M., Iyemori, T., Marchaudon, A., Nosé, M.: Geomagnetic Indices, in Geomagnetic Observations and Models (eds. M. Mandea, M. Korte), IAGA Special Sopron Book Series 5, Springer Science+Business Media B.V., 183-228, doi:10.1007/978-90-481-9858-0_8, 2011.

Line 144: definitions -> determinations? We agree with the reviewer. In this case, it is more correct to use the term 'determinations'.

Line 193: Data is stored not in Excel but in ASCII-Text-files? The data in PANGEA is indeed stored as text files in ASCII codes. We agree with this remark.

Figure 3 caption: Please mention in the figure caption the quiet curve shown in panel (c) We have added to the Fig. 4 (formerly 3ÑĄ) caption explanation that the quiet curve is quiet daily variation Sq.
* * *

---

## Author Comment (AC2) · 19 Feb 2021

We would like to thank the Referee for the comments. They helped us to improve the manuscript.

Lines 80–81 and Figure 2 The text describes 22 and 26 observatories, although Figure 2 shows locations of 41 geomagnetic observatories that may correspond to Table 1. Please indicate locations of the 22 and 26 observatories in Figure 2 or move the statement of "(Fig. 2)" to Line 87. We agree with the reviewer. The 22 and 26 observatories mentioned in the text are not marked among 41 observatories in Figure 2. Therefore, we move the reference to the Figure 2 to Line 89

Lines 99–101 and Figure 3 (a) This line is too simple to understand Figure 3. Needs

more detailed statements. We have made reference here only to the images of historical analogue magnetograms (Fig. 3a, 3b). Figure 3c was given separately under number 4. The link to (Fig. 4) was moved to line 110, where the process of manually determining the index is described. We have made a more detailed caption to fig. 4. (b) Figures 3a and 3b are too small to read labels or necessary information. Please enlarge. In figure caption, "b)" is missed. We enlarged copies of magnetograms (Fig. 3a, 3b) and inserted b). (c) In Figure 3c, the purple bars are drawn only for the second trace (probably the H component), but why? The K index should be determined from both the H and D components (Lines 105–106). The Figure 4 (formerly 3ÑĄ) shows only example, how the amplitudes on one component are determined. This is indicated in the figure caption. (d) What is the date of the magnetograms in Figure 3c? This is just an example. The date doesn't matter. Table 2 Scales 3 and 7 are not adopted to the 41 geomagnetic observatories shown in Table 1. Are these scales needed in this table? We agree that scales 3 and 7 are not needed to describe of observatories considered in this article, and we have removed lines 3 and 7 from Table 2.   Line 145–146 and Figure 5 In Figure 5, there seems no indication of K9 limit. We replaced Figure 6 (formerly 5) with another one with indication of K9 limit.

Lines 163–164 The long-term data are very important. The referee suggests showing plots of the long-term (34–36 year) K index from a few representative observatories. Such plots will interest readers to access the data collection. We plotted K index daily mean values for the period 1958–1992 for three observatories: Heiss Island (HIS), Irkutsk (IRT), and Vostok (VOS). These observatories were selected to include high (HIS) and middle latitudes (IRT) for the northern hemisphere and Antarctica (VOS). This plot is added as figure 8.

[Figure]

[Figure]

**Fig. 1.** Figure 3: Photocopies of analogue magnetograms: a) Petropavlovsk (Paratunka) observatory, 16–17.08.1970, Bobrov variometer; b) Irkutsk (Patrony), 14.07.1960, Yanovsky variometer.

**Fig. 2.** Figure 6: Monthly K index values datasheet for observatory Tbilisi (TFS), Georgia (July 1990), compiled according to the form, proposed by IZMIRAN

**Fig. 3.** Figure 8: K index daily mean values for the period 1958–1992 for three observatories:
a) Heiss Island (HIS); b) Irkutsk (IRT); c) Vostok (VOS).

---

## Author Response (AR1)

**General list of amendments**

*Line 11:*

The word 'solar' changed to lowercase.

*Line 23:*

*K* is given italics.

*Line 48:*

We have replaced the transliteration version in the text with **'Severnyj Polyus'** and the abbreviations **SP** to **NP**. We have also replaced the term 'romanized' with **'transcribed Cyrillic to Latin'**.

*Lines 71–73:*

The sentence was changed to:

"To quantify geomagnetic disturbances caused by the interaction of solar corpuscular radiation with the magnetosphere, processes in the magnetosphere itself, the interaction of the magnetosphere and the ionosphere, as well as processes in the ionosphere itself, the geomagnetic community widely uses geomagnetic activity indices (Lincoln, 1967)".

*Line 89:*

We moved the reference to the Figure 2 from line 83 to line 89.

*Line 102:*

We have replaced **relative to** with **minus.**

*Line 103:*

Reference to the Figure 3 moved to line 103.

*Line 110:*

Reference to the new Figure 4 added to line 110.

*Lines 125–132:*

We have added the following text to the article (line 125–127): "This fact is described in articles (Bartels et al., 1939; Lincoln, 1967). For observatories established after 1938, the lower limit of the amplitude for $K = 9$ is chosen in consultation with the working group on geomagnetic activity indices IAGA (before 1954 IATME) (Lincoln, 1967)".

We have added the following text (lines 127–132):

"An event with $K \leq 2$ is quiet, $K = 2$–$3$ is slightly disturbed, $K = 4$ is disturbed, $K = 5$–$6$ is a magnetic storm, and $K \geq 7$ is a large magnetic storm. This version of the classification is published on the Paratunka Observatory website http://www.ikir.ru/ru/Departments/Paratunka/lfg/txt/k-index-doc.html.

Another classification is given in the work (Menviell et al., 2011): "The modern consensus is that $K = 0$–$2$ correspond to periods of magnetic quietness; $K = 3$–$5$ correspond to periods of moderate geomagnetic activity; $K = 6$–$9$ correspond to periods of intense to very intense geomagnetic activity".

We added the corresponding reference (**line 296**):

Menvielle, M., Iyemori, T., Marchaudon, A., Nosé, M.: Geomagnetic Indices, in Geomagnetic Observations and Models (eds. M. Mandea, M. Korte), IAGA Special Sopron Book Series 5, Springer Science+Business Media B.V., 183-228, doi:10.1007/978-90-481-9858-0_8, 2011.

*Line 151:*

The figure number changed to **Fig. 5a.**

*Line 152:*

The figure number changed to **Fig. 5b.**

*Line 153:*

The figure number changed to **Fig. 6.**

*Line 153:*

We used the word **'determinations'**.

*Line 168:*

The figure number changed to **Fig. 7.**

*Line 172:*

We added the code for observatory 'Vostok' – **'VOS'.**

*Lines 173–175:*

We added a short explanation for the reference to Figure 8:

"As an example, Fig. 8 shows the plots of K index daily mean values for the period 1958–1992 for three observatories – Heiss Island (HIS), Irkutsk (IRT), and Vostok (VOS)".

***Line 204:***

We changed the phrase to "…language and ASCII-Text-files for data storage".

***Figure 3 caption:*** *Please mention in the figure caption the quiet curve shown in panel (c)*

We have added to the Fig. 4 (formerly 3c) caption explanation that the quiet curve is quiet daily variation *Sq.*

***Lines 296–298:***

We added the reference:

Menvielle, M., Iyemori, T., Marchaudon, A., Nosé, M.: Geomagnetic Indices, in Geomagnetic Observations and Models (eds. M. Mandea, M. Korte), IAGA Special Sopron Book Series 5, Springer Science+Business Media B.V., 183-228, doi:10.1007/978-90-481-9858-0_8, 2011.

***Line 325, Table 2***

We have removed lines 3 and 7 from Table 2.

***Line 335 and Figure 3***

We enlarged copies of magnetograms (Fig. 3a, 3b) and inserted b).

***Line 340 and Figure 4***

Former Figure 3c was removed as a separate figure under number 4. The link to (Fig. 4) was moved to line 110, where the process of manually determining the index is described. We have made a more detailed caption to fig. 4.

***Line 341:***

Number of the figure changed from 4 to 5 in the caption.

***Line 346–348 and Figure 6***

We replaced Figure 6 (formerly 5) with another one with indication of $K9$ limit.

Number of the figure changed from 5 to 6 in the caption. The month changed to **'July 1990'**. The date of last access updated.

***Line 350:***

Number of the figure changed from 6 to 7 in the caption.

***Lines 352–353***

We plotted *K* index daily mean values for the period 1958–1992 for three observatories: Heiss Island (HIS), Irkutsk (IRT), and Vostok (VOS). This plot is added as figure 8.

---

## Author Response (AR2)

**General list of amendments**

*Line 322*

The letter K for 'K index' was changed into italics.

*Line 324*

The letter K for 'K index' was changed into italics.

*Figure 4 and Lines 339–343*

We have amended the figure: tags for all three plots added, as well as the scale for K index.

The caption was altered as follows:

Figure 4: A schematic representation of the manual determination of geomagnetic variations' amplitudes for K index calculation. Purple vertical lines correspond to the maximum disturbances in the H component in every 3-hour interval. Similarly, the maximum disturbance amplitudes for the D component are determined. Then the values of the quiet daily variation Sq in each 3-hour interval are subtracted from the amplitudes. The largest value of the two amplitudes obtained for each of the 3-hour intervals using the observatory K index scale is converted into the K index value and adopted as the final one.

*Figure 8*

Figure 8 was altered according to the referee's recommendations.